# Endodontic Regeneration Therapy: Current Strategies and Tissue Engineering Solutions

**DOI:** 10.3390/cells14060422

**Published:** 2025-03-12

**Authors:** Moe Sandar Kyaw, Yuya Kamano, Yoshio Yahata, Toshinori Tanaka, Nobuya Sato, Fusami Toyama, Tomose Noguchi, Marina Saito, Masato Nakano, Futaba Harada, Masahiro Saito

**Affiliations:** Department of Restorative Dentistry, Division of Operative Dentistry, Graduate School of Dentistry, Tohoku University, Sendai 980-8575, Japan; moe.sandar.kyaw.e6@tohoku.ac.jp (M.S.K.); yoshio.yahata.a7@tohoku.ac.jp (Y.Y.); toshinori.tanaka.a8@tohoku.ac.jp (T.T.); nsatossy@h7.dion.ne.jp (N.S.); husami.nishikata.s7@dc.tohoku.ac.jp (F.T.); noguchi.tomose.t1@dc.tohoku.ac.jp (T.N.); saito.marina.p1@dc.tohoku.ac.jp (M.S.); masato.nakano.d8@tohoku.ac.jp (M.N.); futaba.harada.t4@dc.tohoku.ac.jp (F.H.); masahiro.saito.c5@tohoku.ac.jp (M.S.)

**Keywords:** pulp, apical periodontitis, bone regeneration, tissue engineering

## Abstract

With increasing life expectancy and an aging population, the demand for dental treatments that preserve natural teeth has grown significantly. Among these treatments, endodontic therapies for pulpitis and apical periodontitis play a vital role, not only in keeping occlusal function, but also in preventing the exacerbation of systemic diseases. Both pulpitis and apical periodontitis are primarily caused by infections of the oral pathobiont within the root canal, leading to inflammation and destruction of the pulp, apical periodontal tissue, and bone. Standard root canal therapy aims to remove the infection source and facilitate natural tissue healing through the body’s regenerative capacity. However, challenges remain, including limited tooth functionality after complete pulp removal in pulpitis and insufficient recovery of the large bone defect in apical periodontitis. To address these limitations, endodontic regenerative therapies have emerged as promising alternatives. Pulp regeneration therapy seeks to restore the functionality of dental pulp, while bone regeneration therapy aims to repair and regenerate large bone defects affected by apical periodontal tissue.

## 1. Introduction

The dental pulp is a specialized connective tissue located in the core of the tooth and surrounded by a rigid dentin structure. It is composed of fibroblasts, odontoblasts, immune cells, blood vessels, and a dense network of nerves, making it highly responsive to physiological and pathological stimuli [1,2]. The pulp plays a crucial role in maintaining dental health, contributing to dentin formation through odontoblast activity, supplying nutrients to maintain tooth vitality, and transmitting sensory signals [2].

Pulpitis is an inflammatory condition of the dental pulp caused by various irritants, including oral pathobiont invasion due to dental caries, trauma, or restorative procedures. It can be classified as either reversible or irreversible, depending on the extent of damage and inflammation [3]. Reversible pulpitis is characterized by mild inflammation, allowing for recovery when the underlying cause is removed [4]. In contrast, irreversible pulpitis is associated with severe inflammation and tissue damage [5]. If left untreated, pulpitis can progress to more severe pulp-related conditions, such as pulp necrosis, apical periodontitis, and endo-perio lesions, allowing bacterial infections [6].

Apical periodontitis (AP) occurs when bacterial or inflammatory by-products from the root canal spread beyond the apex of the tooth [7]. This process often leads to the destruction of surrounding periapical structures such as bone, periodontal ligament, and the formation of periapical abscesses [8]. If left untreated, the inflammatory process may extend further bone destruction and affect the periodontium, contributing to the development of endo-perio lesions [9].

Endo-perio lesions can be caused by endodontic infections, periodontal diseases, or a combination of both. These lesions are characterized by deep periodontal pockets, periapical radiolucency, tooth mobility, and extensive bone defects, posing significant challenges for healing [10]. Understanding the etiology and progression of these diseases is crucial for effective diagnosis and treatment. Treatments of AP such as root canal therapy (RCT) focus primarily on the removal of the diseased pulp tissue, including the pathological infection and preventing reinfection [11,12]. However, these methods often result in the loss of pulp vitality, which compromises the tooth’s natural defense and repair mechanisms [13]. As a result, the affected tooth becomes more susceptible to fractures, which is one of the main causes of tooth extractions [13]. In addition, patients with systemic diseases such as diabetes, cardiovascular diseases, or autoimmune disorders may develop therapy-resistant apical periodontitis, which increases the risk of large bone defects and further complicates treatment outcomes [14].

Recent advances in endodontic regenerative therapy have created new opportunities to maintain or restore pulp vitality in pulpitis and promote bone regeneration in large defects caused by therapy-resistant conditions or endo-perio lesions [15]. Regenerative approaches include stem cell therapy [16,17], biomaterials, and the modulation of molecular signaling [18]. By focusing on pulp preservation and regeneration, modern dentistry aims to maintain the function and longevity of teeth while minimizing invasive procedures. In addition, the focus on bone regeneration not only includes bone regeneration using stem cells and biomaterials, but also addresses inflammatory conditions in the jawbone, which can be exacerbated by systemic diseases. This review highlights the current state of endodontic regeneration therapy, emphasizing its potential to reshape the future of endodontic research and clinical practice (Figure 1).

This review provides a comprehensive overview of current regenerative procedures and fundamental concepts in endodontics, with a primary focus on stem cell biology in pulp and bone regeneration. Stem cell therapy for pulp regeneration has evolved into a clinically applicable procedure, while bone regeneration strategies using stem cells present a promising future approach for treating apical periodontitis associated with large bone defects, such as endo-perio lesions. The discovery of human bone-forming cells has further clarified the principles of bone regeneration therapy and paved the way for innovative treatments for severe bone loss. These advances improve bone healing and stability, which ultimately improves the long-term success of endodontic treatments.

Building on these findings, this review examines regenerative strategies in dental pulp, particularly cell-based approaches (cell transplantation) using dental pulp stem cells (DPSCs). Additionally, vital pulp therapy, which aims to preserve pulp vitality and maintain physiological function, is discussed. This review also addresses cell-free regenerative approaches (cell homing) that do not require exogenous cell transplantation, as well as emerging technologies for whole-tooth regeneration.

In apical periodontitis, the relationship between systemic inflammation and its impact on healing potential must be considered when assessing the need for bone regeneration. Moreover, strategies for bone regeneration, including the use of bone graft substitutes and tissue engineering technologies, are being actively explored. Bone-forming cell-based therapies, such as the use of skeletal stem cells and immature osteoblast-like cells, show promise for future applications. By integrating recent advances in both pulp and bone regeneration, this review provides a comprehensive perspective on the current state and future directions of regenerative endodontics.

## 2. Current Endodontic Therapy for Promoting Pulp Wound Healing

The treatment strategy for pulpitis depends on its classification—whether it is reversible or irreversible pulpitis—as well as the extent of the inflammation and damage to the pulp. Reversible pulpitis occurs when the pulp becomes partially inflamed, but not permanently damaged. Vital pulp therapy or vital pulp treatment (VPT) focuses on removing the irritant and protecting the pulp to restore normal tooth function [19]. This typically involves the removal of carious tissue and the application of biocompatible pulp-capping agents, such as calcium hydroxide or mineral trioxide aggregate (MTA), to promote healing of the dentin–pulp complex [20]. In contrast, irreversible pulpitis involves severe inflammation and permanent damage to the pulp. Treatment generally requires removing the infected pulp through RCT, where the canals are cleaned, disinfected, and sealed with a biocompatible filling material to prevent reinfection. Recent advances in regenerative endodontic therapy (RET) offer promising alternatives, particularly for young patients with immature teeth [21]. These techniques aim to revitalize the pulp tissue, promote continued root development, and restore the functional properties of the pulp.

### 2.1. Vital Pulp Therapy

VPT is a general term encompassing procedures aimed at preserving exposed dental pulp by managing inflammation and preventing infection. This approach preserves all or part of the pulp tissue and is expected to promote root formation, closure of apical foramen, and an increase in the thickness of the tooth structure, particularly in immature permanent teeth [19]. Recently, the European Society of Endodontology [22] and the American Association of Endodontists [23] have published position statements on VPT, and the pulp preservation approach is now widely accepted as a treatment option. VPT consists of three types of procedures: direct pulp capping, partial pulpectomy, and complete pulpectomy (Figure 2A). In each case, a pulp capping material is applied after the removal of carious or inflamed pulp tissue to facilitate the formation of a dentin bridge or reparative dentin [24]. Traditionally, VPT was indicated for the treatment of trauma or accidental pulp exposure during tooth preparation in young, immature permanent teeth. However, in recent years, VPT has also been successfully used for the management of carious pulp exposures in mature permanent teeth, even in cases where there are clinical signs and symptoms suggestive of irreversible pulpitis [25,26]. This is mainly due to the emergence of MTA, which has high biocompatibility and excellent sealing properties [27,28]. Despite these advancements, histological analysis has not confirmed true pulp regeneration. Therefore, the primary goal of VPT is to promote the healing and repair of pulp tissue, rather than regenerating it to its original state.

### 2.2. Regenerative Endodontics

On the other hand, the regeneration of dental pulp tissue is being explored. The ultimate goals of treatment are the regeneration of pulp tissue, the restoration of sensory nerve response, the formation of secondary dentin, and ultimately the longevity of the tooth. There are three approaches to regenerative endodontic treatment: the cell-free approach, the cell-based approach, and the cell-homing approach [29]. The cell-free approach aims to induce vascularization of the root canal system by inducing the proliferation of soft tissue from periapical tissue or bleeding from the root apex, a process also called revascularization [16].

The first case report [30] describes the treatment of an immature permanent tooth with pulp necrosis caused by a fracture of the central tubercle. Instead of the traditional apexification procedure, which was typically used for such cases, they employed a novel approach. The tooth was treated with root canal irrigation and antimicrobial medication, followed by sealing of the access cavity once soft tissue formation was observed in the canal. Follow-up observation confirmed increased root length, closure of the apical foramen, and obliteration of the pulp chamber. This case marked a significant shift in the treatment of immature teeth with pulp necrosis, and the importance of this treatment is the presence of the apical papilla (Figure 2B) [31,32]. Huang et al. [32] suggested that the healing pattern differs according to the vital state of the apical papilla, which contains stem cells for root development and regeneration. The apical papilla, a stem cell-rich tissue located at the base of developing dental roots, can remain vital or moderately inflamed despite pulp necrosis and apical periodontitis due to collateral circulation near the apex [33]. When comparing cells from inflamed and normal tissues, it was found that the moderately inflamed human apical papilla still maintained vitality and contained stem cells known as SCAP (Stem Cells from the Apical Papilla). SCAP, a type of mesenchymal stem cell, retained their stem cell properties and exhibited enhanced osteogenic and angiogenic potential under inflammatory conditions [34]. However, if the apical papilla becomes necrotic, healing may lead to the infiltration of cementum-like tissue and periodontal ligament into the root canal space [35]. Therefore, the cell-free approach is believed to facilitate the healing and repair of the remaining dental papilla, similar to vital pulp therapy (VPT) [36].

The cell-homing approach is a cell-free regenerative strategy in which the endogenous stem cells are recruited into the root canal through physical or biological stimuli in order to regenerate the dental pulp tissue [16]. In cell homing, effective disinfection of the root canal system and achieving a sterile environment are crucial. Tissue repair and regeneration can only take place in a sterile or highly disinfected microenvironment in which the host’s defense mechanisms do not trigger tissue-destroying proinflammatory processes [37]. In chronic infections, microorganisms penetrate the dentinal tubules and form biofilms. This infected dentin surface prevents cell adhesion and fails to provide structural support for mobilized cells in the main canals [38]. To improve disinfection, current protocols [39] recommend the use of a lower concentration of sodium hypochlorite (1.5–3%), followed by saline or ethylenediaminetetraacetic acid (EDTA). Calcium hydroxide or a triple antibiotic paste (0.1 mg/mL) are recommended as intracanal medicaments. At the second appointment, scheduled 1 to 4 weeks later, 17% EDTA is recommended as an irrigant to further optimize the microenvironment for regeneration.

In cell homing, the blood clot formed by physical stimuli, such as periapical-induced bleeding, is primarily used as a scaffold. However, other scaffolds, such as PRP (platelet-rich plasma), platelet-rich fibrin (PRF), and collagen membranes, have also been utilized. PRP (platelet-rich plasma) is a first-generation platelet concentrate prepared using a double centrifugation process and mixed with anticoagulants [40]. PRF (platelet-rich fibrin), on the other hand, is a second-generation platelet concentrate with a simpler preparation method compared to PRP, requiring only a single round of centrifugation and no addition of anticoagulants or other chemicals. PRF forms a fibrin network that can slowly release growth factors and cytokines, promoting tissue regeneration and accelerating wound healing [41]. Injectable platelet-rich fibrin (i-PRF) has also been utilized, and it is a 100% autologous biomaterial prepared without anticoagulants or additives [42]. Its preparation involves lower centrifugation speeds and shorter durations, allowing for gradual fibrin formation and the creation of a flexible, dynamic hydrogel. The regenerative capabilities of i-PRF and PRP were tested on sixty-two immature maxillary incisors, with or without periapical lesions. After a 12-month follow-up period, no changes were observed in root canal diameter, length, or pulp sensibility as assessed by an electric pulp tester. However, i-PRF showed a significant improvement in apical canal constriction compared to PRP [42].

Collagen membranes have also been investigated as a scaffold material to improve dentin formation in immature teeth. A recent study [43] treated eighty non-vital immature teeth with REPs with and without a collagen membrane (Bio-Gide; Geistlich Pharma AG). The group with the collagen membrane showed a significant increase in dentin deposition in the middle third compared to the blood clot-only group. Moreover, the length of the root increased by 14%, and the thickness of the dentin wall increased by 25% compared to pre-treatment measurements, although there was no significant difference between the groups [43].

In conclusion, the cell-homing approach in regenerative endodontics holds significant promise as an innovative strategy for restoring pulp vitality and function. However, this approach comes with a number of advantages and challenges, as it depends primarily on the recruitment of endogenous stem cells through cell migration mechanisms. Cell-free approaches are effective to a certain extent; they do not regenerate complete dental pulp tissue. Therefore, the development and implementation of cell-based approaches using stem cells are crucial to achieve tissue regeneration and advance the field further.

## 3. Dental Pulp Regeneration Therapy

Cell-based approaches, involving the transplantation of mesenchymal stem cells, have demonstrated successful regeneration of vascularized pulp-like tissue and reinnervation in limited clinical trials [44]. This method offers the potential for more predictable and complete regeneration of pulp and dentin. To understand the regeneration of dental pulp, it is necessary to understand tooth development [45].

### 3.1. Tooth Development

The interaction between the dental mesenchyme and the dental epithelium is central to the tooth development process [46]. The process begins during embryogenesis, with signals from the dental epithelium initiating the condensation of mesenchymal cells derived from neural crest origins. This reciprocal interaction triggers a cascade of signaling pathways, including BMP, FGF, Wnt, and Shh, which regulate tooth morphogenesis and cellular differentiation during these processes [47]. During the bud stage, epithelial cells proliferate into the mesenchyme, giving rise to the dental epithelium and dental mesenchyme, which together form the tooth germ. In the cap stage, the dental epithelium develops into the enamel organ, while the dental mesenchyme differentiates into the dental papilla and dental follicle. During the subsequent bell stage, the dental papilla develops into odontoblasts and pulp tissue, while the dental follicle forms periodontal tissues, including cementum, alveolar bone, and periodontal ligament. Ameloblasts, differentiated from the inner enamel epithelium, and odontoblasts, derived from the dental papilla, form enamel and dentin, respectively, through strictly regulated signaling pathways. These epithelial–mesenchymal interactions not only govern the morphogenesis of the tooth, but also play a critical role in regenerative therapies [48] (Figure 3, upper panel).

Modern tooth regeneration technologies aim to replicate the natural developmental process by utilizing cells derived from the cap stage of the tooth germ. These technologies focus on mimicking the signaling interactions between mesenchyme and epithelium to bioengineer teeth with fully functional structures. Understanding these interactions provides a basis for developing innovative therapies to restore lost or damaged teeth, moving regenerative dentistry closer to clinical applications.

### 3.2. Tissue Engineering Technology of Tooth Regeneration

Bioengineered tooth regeneration technology has been developed, leveraging the principles of tooth development to create fully functional bioengineered teeth [49]. This approach involves the precise manipulation of dental epithelial and dental mesenchymal cells isolated from cap stage mouse tooth germs to mimic the natural processes of odontogenesis. Bioengineered tooth germs have demonstrated successful regeneration of teeth that are capable of integrating into the jawbone with proper physiological functions, including sensation, mastication, and occlusion. A key innovation in this approach is the use of three-dimensional organ germ engineering, in which epithelial and mesenchymal cells are cultivated to re-establish their reciprocal interactions. They then undergo natural development and erupt as fully functional teeth after implantation in the alveolar bone (Figure 3, Lower panel). This technology has also highlighted the potential to restore not only tooth structure, but also periodontal tissue, demonstrating the regenerative capacity of this approach in the treatment of tooth loss. These advances mark a significant step toward the clinical application of bioengineered teeth and represent a promising alternative to traditional dental implants and prosthetics in the future of regenerative dentistry. Although these technologies have demonstrated success in mouse models, questions remain regarding their applicability in human systems, including the generation of human dental mesenchyme and dental epithelium.

### 3.3. Cell-Based Approaches for Pulp Regeneration

Regeneration of the dental pulp tissue involves reconstruction of the pulp dentinal complex, including blood vessels, nerve fibers, and dentin structure, following injury or disease. The process of pulp tissue engineering relies on three critical components: stem cells, biomaterials, and growth factors [50]. Stem cells possess multidirectional differentiation potential and are responsible for tissue healing and regeneration [51]. In particular, dental pulp stem cells (DPSCs) are multipotent mesenchymal stem cells residing in the dental pulp that possess remarkable self-renewal capacity and the ability to differentiate into various cell types, including odontoblasts, osteoblasts, and neurons, making them a promising source for regenerative therapies in dentistry and beyond [52].

Biomaterials can provide a three-dimensional growth space for cells and regulate stem cell function. Growth factors can enhance the regenerative effects and regulatory functions of stem cells [53]. Successful pulp regeneration requires the creation of a suitable reparative environment through the optimal combination of these three components. Resolution of clinical signs and symptoms, further maturation of the root, and neurogenesis are the three outcomes expected using these procedures (Figure 4).

Table 1 summarizes selected articles on pulp regeneration therapy applied in clinical trials. Since animal studies were also excluded from this review, the number of human clinical trials using stem cells was limited. Of the five studies reviewed, two were randomized controlled trials, while three were non-controlled clinical studies. The targeted teeth for pulp regeneration included both immature and mature teeth, with or without apical periodontitis. The intervention in these studies involved a cell-based regenerative approach using various types of stem cells, either autogenic or allogeneic. To ensure a consistent assessment of the results, we included studies with a follow-up period of at least one year. Treatment success was defined by the resolution of infection-related signs and symptoms, revascularization of the pulp, root development, and narrowing of the apical foramen.

The first two studies focused on immature teeth. The first clinical study [54] investigated the use of human exfoliated deciduous dental pulp stem cells (SHEDs) for the treatment of traumatic immature incisors with pulp necrosis. Pulp vitality and dentin formation were assessed after a 12-month follow-up period using laser Doppler flowmetry and CBCT analysis [54]. In the second study [55], allogeneic bone marrow-derived mesenchymal stromal cells were used to treat immature non-vital teeth. After a 12-month observation period, healing and tooth sensitivity were evaluated using electric pulp testing and cold tests [55].

Three studies focused on pulp regeneration in mature teeth. The first study [56] investigated the use of autologous mobilized dental pulp stem cells with granulocyte-colony stimulating factors (G-CSFs) and an atelocollagen sponge, which were implanted into mature single-root canals diagnosed with irreversible pulpitis. Pulp sensitivity and dentin formation were assessed after 12 months using electric pulp testing, MRI signal intensity, and CBCT. The second study [57] investigated the efficacy of allogeneic human umbilical cord mesenchymal stem cells (UC-MSCs) in combination with platelet-poor plasma in the treatment of mature teeth with pulp necrosis or apical periodontitis (n = 18). Pulp vitality was evaluated using electric pulp testing and laser Doppler flowmetry after 12 months [57]. The final study [58] explored direct pulp tissue transplantation, a technique that involves the direct transplantation of dental pulp stem cells, bypassing complex in vitro procedures. In this study, six mature necrotic permanent teeth were treated with minced pulp tissue grafts following apical-induced bleeding. After a follow-up period ranging from 19 to 42 months, periapical lesions were assessed both clinically and radiographically, and pulp sensitivity was evaluated using electric pulp testing [58].

The five studies summarized above have shown promising results in both immature and mature teeth, demonstrating the potential for pulp healing and revitalization. Among the 70 teeth examined in the five studies, 65 were diagnosed with pulp necrosis or periodontitis. After a 12-month follow-up, all 65 teeth (100%) showed a reduction in periapical lesions indicating a high success incidence. The five teeth from Nakashima et al. (2017) [56] were excluded due to the diagnosis of irreversible pulpitis. Regarding pulp vitality, of the 70 teeth tested, 64 (91.4%) responded positively to electric pulp testing, while six teeth (8.6%) showed no response. It is noteworthy that five of the non-responding teeth (7.1%) were from the study by Kim et al., which involved the direct transplantation of minced pulp tissue.

To further evaluate the viability of the pulp, two studies [54,57] utilized laser Doppler flowmetry and reported an increase in perfusion units between 6.39 and 17.5. In one study [56], the signal intensity of the regenerated pulp compared to the normal pulp was examined using MRI. It was found that all five teeth examined had a signal intensity comparable to that of normal teeth, suggesting complete pulp regeneration after 24 months. In the remaining two studies [55,58], neither MRI nor laser Doppler flowmetry was used to assess viability.

Regarding root dentin deposition and apex closure, a study on immature teeth byGomez et al. reported that all 15 teeth demonstrated apical closure and dentin deposition. Xuan et al. [54] observed a mean decrease in root length of 2.64 ± 0.17 mm over 12 months. Nakashima et al. [56] reported dentin deposition in three out of five mature teeth. However, dentin deposition was not explicitly mentioned in the other two studies.

As far as the safety of stem cell implantation is concerned, urine and blood tests were carried out in two studies. These studies analyzed immune cell profiles, including CD4+ T lymphocytes, CD8+ T lymphocytes, B lymphocytes, and natural killer (NK) cells. In addition, key biochemical markers such as blood bilirubin, lactate dehydrogenase, creatine phosphokinase, uric acid, antistreptolysin O, C-reactive protein, rheumatoid factor, immunoglobulin A (IgA), IgG, IgM, C3, and C4, and myocardial enzymes were within normal ranges, indicating that there was no systemic immune response or organ dysfunction following stem cell therapy. The remaining three studies did not mention specific laboratory assessments, but reported no systemic or local adverse effects related to the procedure.

Although stem cell-based therapy has been reported to be safe, allogeneic therapy remains susceptible to immune rejection. Since dental pulp stem cells (DPSCs) and mesenchymal stem cells (MSCs) are not immune-privileged, they can be recognized and rejected by both innate and adaptive immune responses [59]. This immune recognition may compromise treatment outcomes and lead to reduced longevity and viability of regenerated pulp tissue [60].

Furthermore, the American Association of Endodontists (AAE) (2021) and the European Society of Endodontology (ESE) (2016) [39,61,62] have not yet recommended stem cell-based pulp regeneration for routine clinical practice. Despite its adoption in clinical trials, several challenges hinder the feasibility of this therapy. These include difficulties in standardizing protocols for stem cell isolation and expansion, limited access to Good Manufacturing Practice (GMP) facilities, and high costs associated with production and application. Addressing these limitations is essential for the successful translation of pulp regeneration therapies into mainstream endodontic practice.

Recent advances in regenerative technology include cell sheets, spheroids, organoids, and 3D bioprinting [63]. Cell sheets provide a scaffold-free structure that forms high-density sheet structures out of cells and their extracellular matrix [64]. Spheroids are dense 3D cell aggregates with enhanced cell–cell and cell–matrix interactions which create a potent secretome that promotes angiogenesis and tissue repair [65,66]. Dental pulp organoids have emerged as promising models by co-culturing DPSCs with endothelial cells in a 3D environment to create pre-vascularized organoids. Additionally, layered scaffolds, designed using computer-aided design (CAD), facilitate periodontal and dental tissue regeneration by incorporating stem cells, biomaterials, and growth factors [67,68]. These technologies can help overcome challenges in stem cell therapy, such as poor cell retention and rapid cell death, while also advancing personalized dental treatments.

**Table 1 cells-14-00422-t001:** Summary of Completed and Published Clinical Trials on Stem Cell Transplantation.

Author	Transplant Group	Origin of the Cells	Type of Transplantation	Outcome
Nakashima et al., 2017 [56]	Irreversible pulpitis of mature single root canal (n = 5)	Autologous mobilized dental pulp stem cells (mobilized DPSCs) from permanent dental pulp	Mobilized DPSCs + granulocyte colony-stimulating factor (G-CSF) + atelocollagen sponge	Pulp sensibility was restored in four out of five patients via EPT. MRI signal intensity was comparable between test and control teeth, while CBCT revealed lateral dentin formation in three out of five cases.
Xuan et al., 2018 [54]	Traumatic immature incisor with pulp necrosis (n = 26)	Autologous stem cells from human exfoliated deciduous dental pulp (SHEDs)	SHED (1 × 10^8^ cells) were implanted into injured teeth	Dental pulp tissue regeneration after 12 months follow-up. Increase vascular formation by laser Doppler flowmetry. Increased root length and reduced the width of the apical foramen by CBCT analysis.
Brizuela et al., 2020 [57]	Mature teeth with pulp necrosis or apical periodontitis (n = 18)	Allogenic human umbilical cord mesenchymal stem cells (UC-MSCs)	Encapsulated UC-MSCs + platelet-poor plasma	Test groups showed increased positive pulp responses and perfusion units via laser Doppler flowmetry at 12 months follow-up.
Gomez-Sosa et al., 2024 [55]	Immature teeth with pulp necrosis or apical periodontitis (n = 15)	Allogenic bone marrow mesenchymal stem cells (BM-MSCs)	BM-MSCs + autologous platelet-rich plasma (PRP)	Clinical and radiographic evaluation of the treated teeth showed periapical lesion healing, sensitivity to cold and electricity, decreased width of the apical foramen, and mineralization within the canal space.
Kim et al., 2025 [58]	Mature teeth with pulp necrosis or apical periodontitis (n = 6)	Autologous dental pulp stem cells from permanent dental pulp	Autologous minced pulp grafting in teeth	After 19 to 42 months, periapical lesions were resolved in all teeth, both clinically and radiographically, with one tooth regaining sensibility, while three out of six teeth showed calcification.

## 4. Bone Regeneration Therapy for Apical Periodontitis

Apical periodontitis results in inflammatory bone resorption and can lead to complications such as abscess formation, cyst formation, or systemic infections. Treatment generally focuses on eliminating the source of infection, either by RCT to clean and seal the root canal system, or by endodontic surgery to remove bone lesions. However, in endo-perio lesions with extensive bone defects, tooth extraction may be necessary due to insufficient bone support. In such instances, a combination of RCT and bone regenerative therapies plays a critical role in addressing progressive bone destruction and restoring the health of periapical tissues. To overcome these challenges, it is essential to develop effective bone tissue engineering strategies that integrate insights from the pathology of apical periodontitis, materials science, and stem cell biology.

### 4.1. Pathology of Apical Periodontitis

The formation of apical periodontitis is believed to occur by the following mechanisms: (1) Microbial infection within the root canal or around the root apex, mechanical damage to the physiological root apex, or chemical injury can trigger an acute inflammatory response in the apical tissue. (2) This leads to the secretion of inflammatory cytokines, resulting in increased permeability of the surrounding capillaries. (3) As inflammation progresses, immune cells such as macrophages and lymphocytes migrate to the site of the lesion. (4) These immune cells further release proinflammatory mediators that activate osteoclasts, leading to progressive bone resorption around the root apex. If the infection persists, the acute inflammation progresses into a chronic state, in which macrophages become the dominant immune cells within the lesion. In response, the host initiates a repair process characterized by the production of new cells, blood vessels, and connective tissue fibers, leading to the formation of granulation tissue (Figure 5).

These inflammatory reactions are believed to be resolved by eliminating the source of infection [69]. However, in older patients or patients with pre-existing conditions, inflammation at the root apex may persist even after removal of the source of infection, leading to persistent tissue destruction and treatment-resistant conditions. Recent studies have shown a correlation between apical periodontitis and systemic disease, suggesting that inflammation in other organs influences the healing potential of apical periodontitis [70]. Therefore, it is important to understand the relationship between apical periodontitis and systemic diseases (Figure 6). Inflammation and alterations in the immune response in systemic diseases can be associated with genetic polymorphisms [71]. These polymorphisms subsequently affect the functions of key cells involved in inflammation and healing, including macrophages, neutrophils, lymphocytes, fibroblasts, and osteoblasts. Several studies have identified associations between specific genetic polymorphisms and the development of treatment-resistant apical periodontitis [72].

In particular, polymorphisms in genes relates to bone metabolism, such as the genes that regulate osteoclast differentiation (RANK and RANKL) [73]. Additionally, single nucleotide polymorphisms in genes related to bone formation, such as BMP2, BMP4, SMAD6, and RUNX2, have been associated with treatment-resistant apical periodontitis [74]. These findings suggest that therapeutic strategies aimed at reducing systemic disease-related inflammatory responses and promoting bone regeneration play a crucial role in the management of apical periodontitis. This approach is particularly important in cases with large bone defects, such as through-and-through lesions or those larger than 5 mm.

### 4.2. Current Approach for Promoting Bone Healing of Apical Periodontitis

The primary approach in the treatment of apical periodontitis is to eliminate the source of infection, especially the bacteria, that can cause inflammatory bone destruction around the root apex. However, the complexity of the root canal system and the host’s immune system can sometimes interfere with the healing process, leading to treatment failure and continued apical bone destruction. Root-end microsurgery has emerged as a reliable treatment option with high success rates [75,76]. This procedure involves the use of operating microscopes and advanced surgical instruments, such as ultrasonic tips for conservative root-end preparation, micromirrors, and biocompatible root-end filling materials. In this procedure, a small osteotomy and root-end resection are performed to remove infected tissue from the area of bone resorption. The healing process after root-end microsurgery can be lengthy, and take between 1 and 6 years, with poor healing outcomes, especially in larger lesions (>5 mm) [77,78] and through-and-through lesions [79]. Therefore, the resulting cavity can either heal naturally or be filled with various synthetic materials to improve bone regeneration. However, synthetic bone substitutes may have limitations such as bio-incompatibility and interference with post-surgical evaluation. To overcome these limitations and improve healing in cases of severe bone resorption, various techniques have been explored, including guided tissue regeneration [80] and the use of platelet-rich plasma (PRP) [81], which contains beneficial growth factors and anti-inflammatory compounds.

A newer alternative, concentrated growth factor (CGF), represents an advancement over PRP. It contains an equivalent or higher amount of growth factors [53] and has a protective gel-like structure. CGF is obtained from the patient’s own blood, which avoids the risk of cross-contamination and compatibility issues [82]. Several studies have reported on the successful use of CGF in root-end microsurgery [83,84,85]. A multicenter, randomized controlled trial conducted at four dental universities demonstrated that CGF significantly improved lesion volume reduction, achieving a rate of 75.6% compared to 61.0% in the control group [85]. Therefore, a combined approach that focuses on reducing infection and promoting bone regeneration could be particularly beneficial for patients with systemic diseases, who often experience slower healing (Figure 7).

## 5. Bone Tissue Engineering of Apical Periodontitis

### 5.1. Bone Substitute Material

Regenerative technologies and bone graft materials are widely used in endodontic surgery to create a favorable microenvironment for tissue and bone regeneration [86]. The most commonly used bone substitutes in endodontic surgery are calcium phosphate, hydroxyapatite, and demineralized freeze-dried bone allograft (DFDBA) material [87,88,89]. A meta-analysis showed that bovine hydroxyapatite could lead to a better outcome after endodontic procedures, especially in medium or large periapical lesions. Furthermore, synthetic granular bone grafts, including hydroxyapatite (HA), β-tricalcium phosphate (β-TCP), and calcium sulfate, have been used to treat bone defects, and have proven their efficacy.

Hydroxyapatite (HA), the primary inorganic component of bone and teeth, has been extensively studied for clinical applications since the 1970s [90]. This calcium phosphate material possesses osteophilic, osteoconductive, and osteointegrative properties, allowing it to bond directly to bone through the natural bone remodeling process [91]. Although HA exhibits osteoconductive properties and excellent biocompatibility, it requires a longer period of time for complete bone substitution. β-tricalcium phosphate (β-TCP) is a widely used and highly effective synthetic bone graft substitute, known for its osteoconductive properties and osteoinductive potential. These properties, along with its cell-mediated resorption, contribute to the complete regeneration of bone defects [92]. Various bone substitutes, including β-TCP, have already been implemented in clinical practice [93], but further development of bone substitute materials that enable faster and more reliable bone healing is expected. Calcium sulfate (CS), which has been introduced as a filler material, has been reported to be biocompatible, biodegradable, osteoconductive, angiogenic, and hemostatic. Its natural abundance and widespread use in both industrial and medical applications make CS a highly relevant material in various technologies [94]. However, CS presents several limitations in terms of regenerative properties, including a rapid resorption rate, a large surface area, and limited regenerative capacity [95]. However, the bone-healing properties of these substitute materials remain a challenge.

To address this limitation, a three-dimensional bone graft was developed by combining granular ceramic bone grafts with collagen fibers. This innovation has made it possible to achieve sufficient regeneration in small bone defects using these bone grafts alone. A bone substitute with a porous three-dimensional structure, primarily composed of collagen and ceramic materials, has been approved in Japan. One such material is octacalcium phosphate (OCP) in combination with atelocollagen (OCP/collagen) [96,97,98,99]. The bone regeneration potential of this material was evaluated by transplanting it into a large bone defect after apical root resection and periapical cyst removal [100]. Successful bone healing was observed after six months. These findings emphasize the effectiveness of three-dimensional structured bone grafting materials in facilitating bone regeneration during endodontic surgery.

Three-dimensional bone substitutes are also available as hydroxyapatite (HAp) and collagen (Col) composites, synthesized by a simultaneous titration co-precipitation process using Ca (OH)_2_, H_3_PO_4_, and porcine atelocollagen. Such a material, which is marketed under the name Refit-Dental, has a self-organized nanostructure like natural bone, which is formed by chemical interactions between HAp and Col [101,102,103]. The consolidated composite, which was subjected to a cold isostatic pressure of 200 MPa, exhibited approximately one-fourth of the mechanical strength of the bone. Additionally, it demonstrated biological properties comparable to those of grafted bone. It was resorbed by phagocytosis by osteoclast-like cells while guiding osteoblasts to facilitate new bone formation in the surrounding area [104]. Another synthetic bioresorbable bone void filler, ReBOSSIS-85 (RB-85), was developed for the repair of bone defects, and has similar handling properties to glass wool or cotton balls [105]. ReBOSSIS has been successfully used in the harvesting of trabecular bone grafts from the calcaneus, and has achieved satisfactory clinical results [106].

Autologous bones have traditionally been considered the gold standard in bone grafting due to their high regenerative capacity and proven efficacy. However, a major challenge when using autologous bone for large bone defects is the invasive nature of the procedure at the donor site, where the pelvic bone and surrounding anatomical structures are significantly disturbed. Freeze-dried bone and demineralized freeze-dried bone from human cadavers have been shown to be effective in promoting new bone formation. However, they are associated with immunological and ethical concerns, as well as the risk of viral infection [107]. Bovine-derived xenogeneic bone consists of carbonate-containing apatite, which shares the same primary component as natural bone, giving it high biocompatibility. However, the success of bone healing with this material is influenced by the experimental conditions [108,109,110].

### 5.2. Application of Skeletal Stem Cells as Bone Forming Cells

Bone regeneration is a lengthy process, typically taking six months to a year, and it is well-established that the structure of bone substitute materials can collapse before sufficient bone formation has occurred [106,107]. These limitations may hinder bone healing in the case of vertical bone regeneration. To overcome this challenge, research has focused on combining bone substitute materials with cells that have osteogenic potential, with the aim of promoting bone regeneration before the structure of the substitute material collapses [108].

Traditionally, mesenchymal stem cells (MSCs) from bone marrow or adipose tissue are used for bone regeneration. However, clinical studies combining MSCs with bone substitutes have emphasized their anti-inflammatory and immunotolerant properties, but they have not consistently shown a direct bone-forming effect. Currently, there are no regenerative medicine products that effectively repair large bone defects or reliably promote new bone formation [111].

It has recently been recognized that human skeletal stem cells (SSCs) are a crucial contributor to bone development and regeneration. These SSCs are characterized by the markers Lin−PDPN+CD146−CD73+CD164+. SSCs are tissue-resident stem cells with multipotent differentiation potential within the skeletal system. They have been identified at various anatomical sites, including the periosteum, skull, vertebrae, and femur. These cells have the capacity for self-renewal and can differentiate into osteogenic lineages [107,108]. Considerable efforts have been made to identify the cellular origins of non-hematopoietic lineages in skeletal tissues, such as bone, cartilage, vascular endothelium, and stroma. However, the lack of reliable cell surface markers, standardized tissue dissociation protocols, and functional stem cell assays has hindered the identification of presumptive stem cells within the human skeleton [112,113,114,115,116]. Recently, a research group succeeded in isolating SSCs from the growth plate of a 17-week-old human fetal femur [117]. Moreover, SSCs have been reported to induce endochondral ossification when transplanted under the renal capsule of immunodeficient mice [109,110,117].

Notably, SSCs have also been discovered in the human mandible. Recent studies have demonstrated that SSCs isolated from adult mandibles possess regenerative capacity when combined with gelatin-based scaffolds, resulting in successful bone regeneration in a rat mandibular defect model [118]. These findings emphasize the promising potential of SSCs in improving and enhancing bone regeneration therapies, especially for challenging anatomical areas like the mandible (Figure 8).

In a recent study [119], human immature alveolar bone osteoblast-like cells (HAOB) were successfully isolated from the jawbone during wisdom tooth extraction. Using a unique enzymatic digestion method, these cells can be isolated, cultured, and used for regeneration purposes. Their bone-forming potential has been observed in an autologous transplantation model [119].

Previous studies have also demonstrated the effectiveness of vertical bone formation using mouse calvaria-derived immature osteoblast-like cells (MCOB), which were isolated and cultured from the mouse calvaria bone using a method similar to HAOB. When combined with a polylactic acid scaffold and after autologous transplantation into a mouse jawbone defect model, these cells successfully regenerated bone with a strength comparable to that of natural bone [117].

Many current cell transplantation treatments for bone regeneration face challenges due to the limited osteogenic potential of the transplanted cells. A promising solution to these problems is jawbone-derived skeletal stem cells (SSCs) with high osteogenic potential. However, further advancements in bone tissue engineering are necessary to effectively integrate osteogenic cells, such as SSCs or HAOB, with suitable scaffolding materials. Additionally, the establishment of a reliable grafting technique for SSCs could facilitate their combination with three-dimensional bone substitute materials. This approach holds great potential for enhancing early healing in apical periodontitis with large bone defects (Figure 9).

## 6. Conclusions

Endodontic regeneration therapy holds significant potential for restoring dental pulp and bone while reducing the prevalence of chronic dental infections, such as apical periodontitis (AP). Both cell-based and cell-free approaches have shown promise in dental pulp regeneration, with studies reporting positive outcomes for both allogeneic and autologous stem cell transplantation. In addition, cell transplantation has demonstrated potential in mature teeth, suggesting that it may be a viable alternative in certain cases. However, long-term studies are still needed to fully evaluate the viability and efficacy of these approaches. Future research should focus on improving long-term stability, optimizing immunomodulation, and standardizing protocols for stem cell isolation and application. Although the efficacy of cell-based therapy for pulp regeneration has been demonstrated, the precise mechanisms by which the transplanted cells promote pulp regeneration and root development remain unclear. This could be due to either the facilitation of wound healing in the periapical tissue through immune tolerance or the regeneration of pulp tissue through tissue reorganization. To address this issue, it is crucial to identify suitable cells for pulp regeneration and to determine whether autologous or allogeneic transplantation is more effective for reliable cell therapy. Further research and clinical trials are essential to addressing these limitations, with an emphasis on developing combination therapies and establishing cost-effective strategies.

Refractory cases of apical periodontitis are often accompanied by large bone defects. Although bone grafting materials with a porous three-dimensional structure, mainly consisting of collagen and ceramic materials, have been developed, the treatment of large bone defects that require vertical bone formation without support from the surrounding bone remains a major challenge. To address this problem, the ability of these materials to regenerate bones must be improved beyond their inherent properties.

One of the major hurdles in bone tissue engineering using materials and cells has been the inability to demonstrate a clear superiority in the bone-forming ability of transplanted cells. To overcome this, techniques need to be developed for the clinical application of skeletal stem cells (SSCs) that have an intrinsic bone-forming ability. By combining SSCs with bone fillers that have a three-dimensional structure, it may become possible to treat large bone defects and manage apical periodontitis with a high degree of predictability.

The regeneration of large bone defects has long been a critical challenge in dentistry. Successfully addressing this issue through advanced bone tissue engineering could significantly improve a patient’s quality of life and contribute to extending healthy life expectancy. As a first step, our focus is on developing bone regeneration techniques for three-wall defects with supporting bone. Once this goal is achieved, we aim to extend our efforts to horizontal bone defects without surrounding bone support.

## Figures and Tables

**Figure 1 cells-14-00422-f001:**
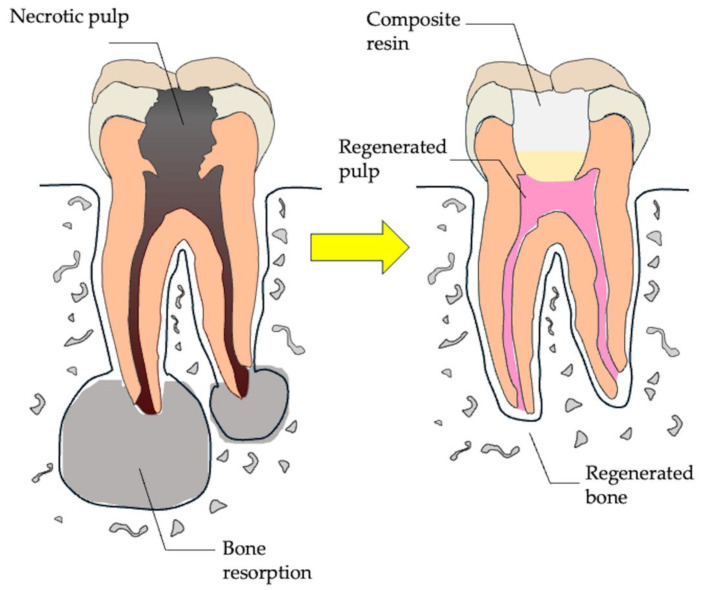
Regeneration therapy for pulpitis and apical periodontitis. Necrotic dental pulp can be treated through pulp regeneration, characterized by reconstruction of the pulpodentinal complex via angiogenesis, neurogenesis, and dentin formation, resolving bone resorption and restoring physiological tooth functions.

**Figure 2 cells-14-00422-f002:**
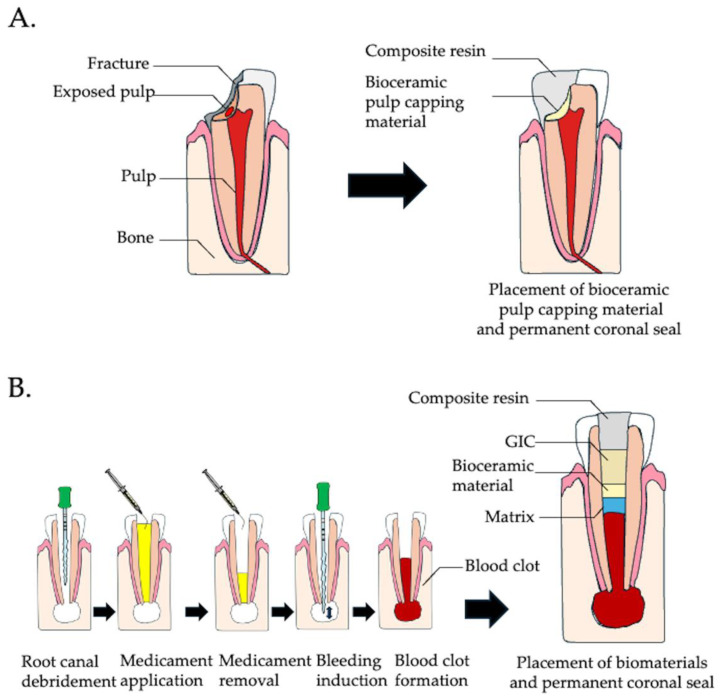
Current regeneration therapy in endodontics. (**A**) Vital pulp therapy: The diseased or inflamed pulp is removed, and a biocompatible pulp-capping material is applied to promote healing and protect the pulp. (**B**) Regenerative endodontic procedure for an immature tooth with pulp necrosis or apical periodontitis. The procedure begins with intracanal debridement followed by the application of a medicament. After the medicament is removed, a file is used to induce a blood clot within the canal, facilitating the migration of endogenous cells into the root canal. Once the blood clot is established, a permanent coronal seal is placed to protect and isolate the regenerative site.

**Figure 3 cells-14-00422-f003:**
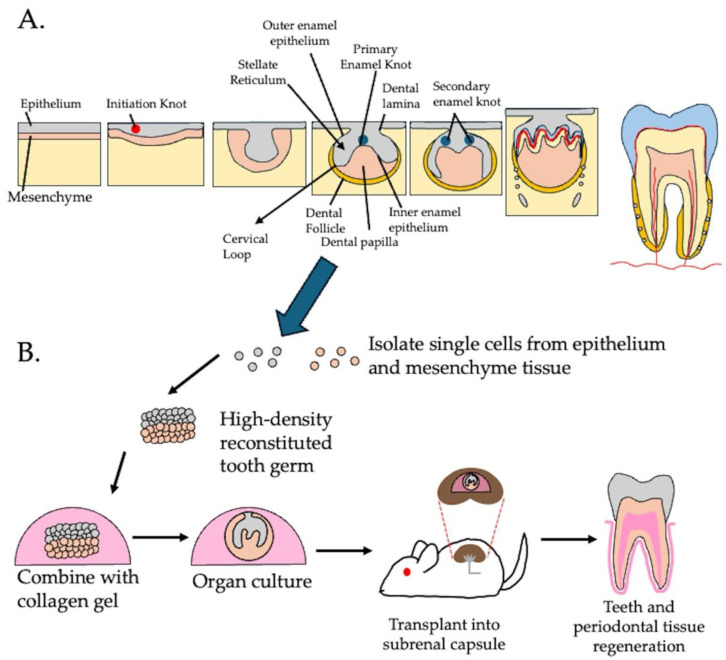
Bioengineered tooth germ technology based on tooth germ development. (**A**) Stages of tooth development; dental placode formation, bud stage, cap stage and bell stage. Following the bell stage, the dental follicle and enamel organ continue to develop and differentiate into a mature tooth. (**B**) Bioengineered tooth germ technology based on reconstruction of dental mesenchyme and dental epithelium.

**Figure 4 cells-14-00422-f004:**
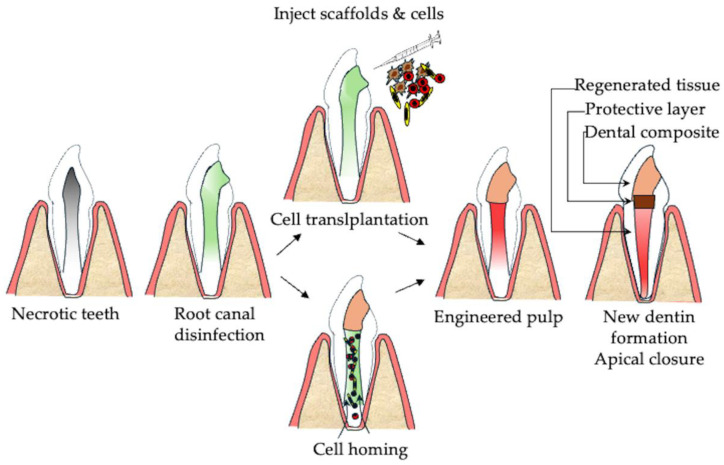
Schematic representation of the regeneration of irreversibly diseased dental pulp tissue using two methods. Cell transplantation, involving the direct injection of stem cells and scaffolds into the root canal; and cell homing, which promotes periapical bleeding to encourage the migration of endogenous cells into the root canal.

**Figure 5 cells-14-00422-f005:**
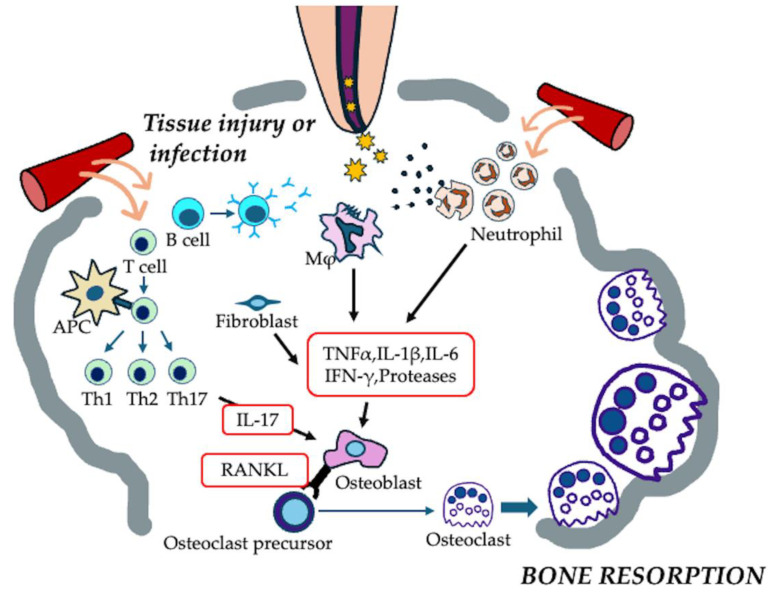
Pathogenesis of apical periodontitis. Microbial infection or tissue injury within the root canal system triggers an acute inflammatory response, leading to the recruitment of immune cells, primarily neutrophils, macrophages, and lymphocytes. These immune cells release pro-inflammatory cytokines, including TNF-α, IL-1β, IL-6, IL-17, and RANKL, which stimulate osteoclast formation and activation, ultimately resulting in bone resorption at the root apex.

**Figure 6 cells-14-00422-f006:**
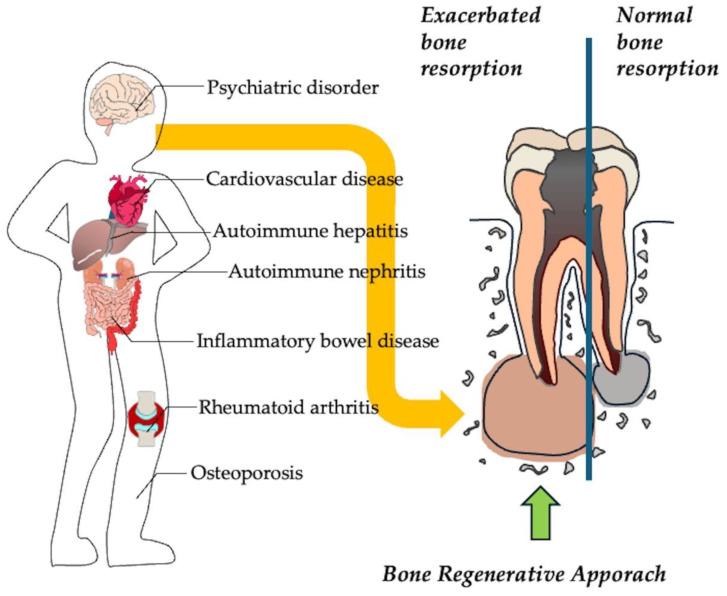
Apical periodontitis accompanied with systemic disease exacerbate bone destruction. Systemic diseases alter immune responses, affecting macrophages, neutrophils, lymphocytes, fibroblasts, and osteoblasts, and subsequently increase pro-inflammatory cytokines, enhancing osteoclast activity and increase bone loss in the periapical region.

**Figure 7 cells-14-00422-f007:**
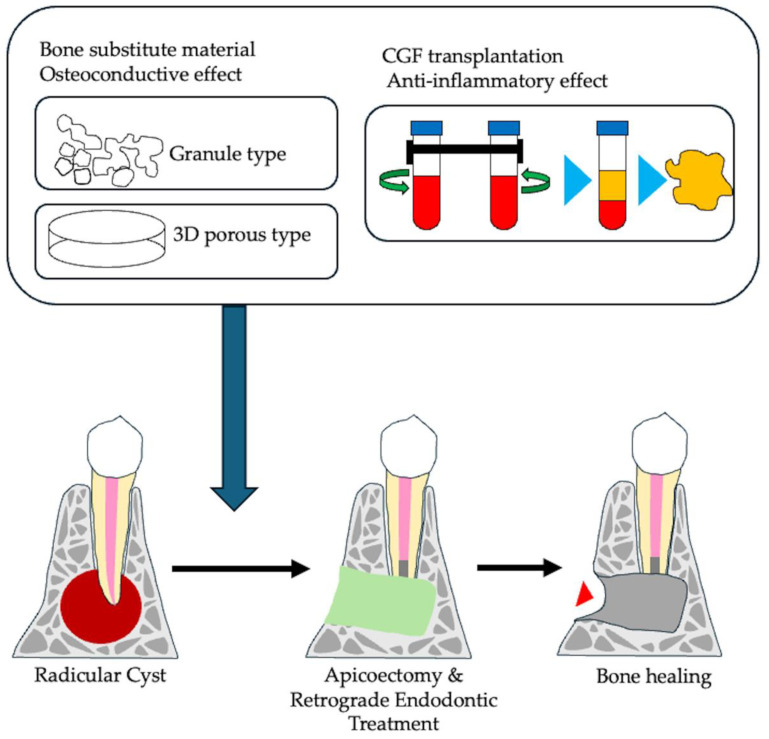
Strategy of apical surgery using tissue engineered technology. A small osteotomy and root end resection was performed to remove infected tissue from the bone resorption area. The resulting cavity can be left to heal naturally or filled with various regenerative materials. Synthetic bone substitute materials, platelet-rich plasma (PRP), or concentrated growth factor (CGF), which contain beneficial growth factors and anti-inflammatory compounds can be used to promote regeneration.

**Figure 8 cells-14-00422-f008:**
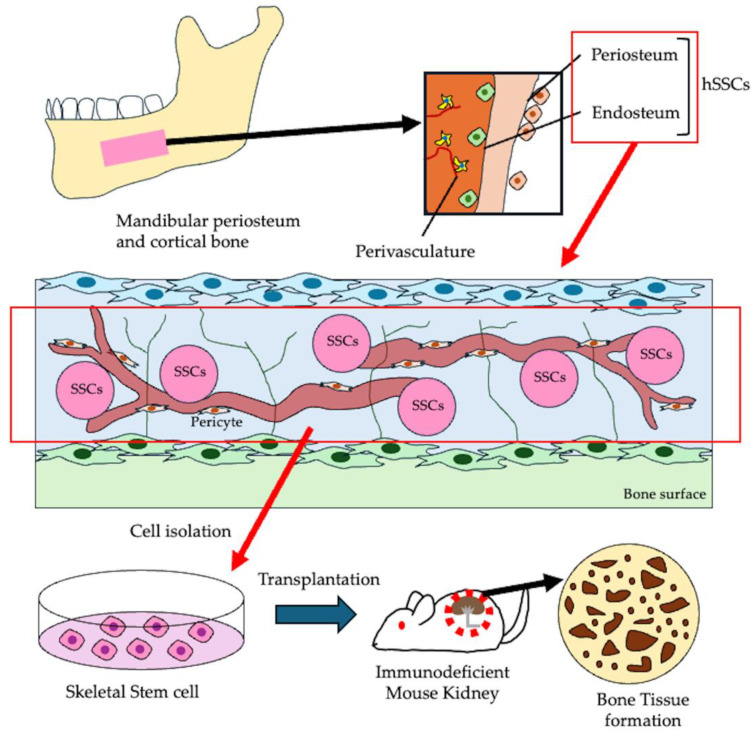
Skeletal stem cell served as bone forming cells. Skeletal stem cells (SSCs) primarily derived from the periosteum and endosteum possess significant regenerative potential, as well as the self-renewal and differentiation abilities required for bone regeneration therapies. These cells can be isolated from the mandibular periosteum and utilized for bone regeneration.

**Figure 9 cells-14-00422-f009:**
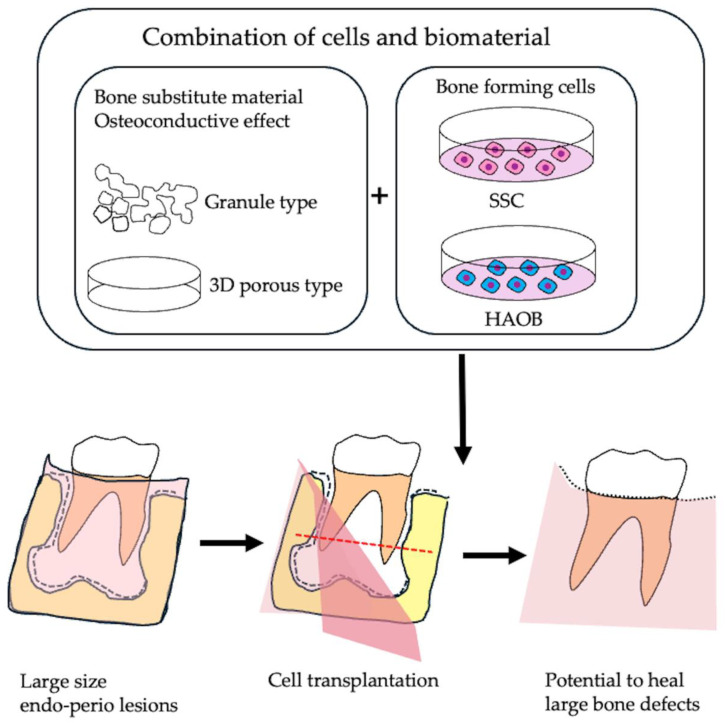
Bone tissue engineering using bone forming cells. SSCs or HAOB with strong osteogenic potential can be combined with granular or three-dimensional bone substitute materials and transplanted into large bone defects, such as endo-perio lesions, to promote early healing.

## Data Availability

Data sharing is not applicable to this article.

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
