# Peer review of "Endodontic Regeneration Therapy: Current Strategies and Tissue Engineering Solutions"

_cells, 2025, doi:10.3390/cells14060422_

Round 1

Reviewer 1 Report

Comments and Suggestions for Authors

0. In the 'Introduction' section, clearly stating the purpose and context of the manuscript would enhance readers' understanding.

1. Recommend adding references in "2.2. Regenerative endodontics" Section, that report on the preservation of stemness and regenerative characteristics of apical papilla-derived cells from immature teeth, particularly in the context of pulpal and periapical pathosis, and under inflammatory conditions.

2. Figure 9 needs revision: overlapped images

3. Figure 6 requires revision: The statement regarding the relationship between apical periodontitis and systemic diseases is too exaggerated. Additionally, the term 'regenerative endodontic treatment' (RET) refers specifically to therapies for promoting healing and root development in immature teeth, rather than approaches aimed at regenerating apical bone. Changing the term would help prevent potential misunderstandings for readers.

Author Response

We would like to thank Reviewer 1 for the constructive feedback on our manuscript. We have carefully addressed each comment point by point and made the necessary revisions to the manuscript accordingly.

COMMENT 0⁠- ⁠In the 'Introduction' section, clearly stating the purpose and context of the manuscript would enhance readers' understanding.

ANSWER- Thank you for your valuable suggestion. We have revised the text of the introduction to ensure the overview of the review article.

Page 3, Line no 69

This review provides a comprehensive overview of current regenerative procedures and fundamental concepts in endodontics, with a primary focus on stem cell biology in pulp and bone regeneration. Stem cell therapy for pulp regeneration has evolved into a clinically applicable procedure, while bone regeneration strategies using stem cells present a promising future approach for treating apical periodontitis associated with large bone defects, such as endo-perio lesions. The discovery of human bone-forming cells has further clarified the principles of bone regeneration therapy and paved the way for innovative treatments for severe bone loss. These advances improve bone healing and stability, which ultimately improves the long-term success of endodontic treatments. 

Building on these findings, this review examines regenerative strategies in dental pulp, particularly cell-based approaches (cell transplantation) using dental pulp stem cells (DPSCs). Additionally, vital pulp therapy, which aims to preserve pulp vitality and maintain physiological function, is discussed. This review also addresses cell-free regenerative approaches (cell homing) that do not require exogenous cell transplantation, as well as emerging technologies for whole-tooth regeneration. 

In apical periodontitis, the relationship to systemic inflammation and its impact on healing potential must be considered when assessing the need for bone regeneration. Moreover, strategies for bone regeneration, including the use of bone graft substitutes and tissue engineering technologies, are being actively explored. Bone-forming cell-based therapies, such as the use of skeletal stem cells and immature osteoblast-like cells, show promise for future applications. By integrating recent advances in both pulp and bone regeneration, this review provides a comprehensive perspective on the current state and future directions of regenerative endodontics. 

COMMENT 1.⁠ ⁠Recommend adding references in "2.2. Regenerative endodontics" Section, that reports on the preservation of stemness and regenerative characteristics of apical papilla-derived cells from immature teeth, particularly in the context of pulpal and periapical pathosis, and under inflammatory conditions.

ANSWER: Thank you very much for your valuable suggestions. We have added information and references regarding the apical papilla derived cells.

Page 5, Line 148

The apical papilla can remain vital and moderately inflamed despite pulp necrosis and apical periodontitis due to collateral circulation near the apex [34]. The apical papilla can remain vital or moderately inflamed despite pulp necrosis and apical periodontitis due to collateral circulation near the apex [34]. When comparing cells from inflamed and normal tissues, it was found that the moderately inflamed human apical papilla still maintained vitality and contained stem cells known as SCAP (Stem Cells from the Apical Papilla). SCAP, a type of mesenchymal stem cell, retained their stem cell properties and exhibited enhanced osteogenic and angiogenic potential under inflammatory conditions [35]. However, if the apical papilla becomes necrotic, healing may lead to the infiltration of cementum-like tissue and periodontal ligament into the root canal space [36]. Therefore, the cell-free approach is believed to facilitate healing and repair of the remaining dental papilla, similar to vital pulp therapy (VPT) [37].

COMMENT 2.⁠ ⁠Figure 9 needs revision: overlapped images

ANSWER: Thank you very much for point out the important information and we sincerely apologize for the overlapped images. We have revised the figure 9 accordingly.

COMMENT 3.⁠ ⁠Figure 6 requires revision: The statement regarding the relationship between apical periodontitis and systemic diseases is too exaggerated. Additionally, the term 'regenerative endodontic treatment' (RET) refers specifically to therapies for promoting healing and root development in immature teeth, rather than approaches aimed at regenerating apical bone. Changing the term would help prevent potential misunderstandings for readers.

ANSWER: Thank you very much for pointing out this crucial detail. We have revised the statement regarding the relationship between the apical periodontitis and systemic diseases. We have also revised the term 'regenerative endodontic treatment' to 'bone regenerative approach'.

Figure 6

Fig. 6 Apical periodontitis is associated with various systemic diseases, including autoimmune diseases (autoimmune nephritis, autoimmune hepatitis, inflammatory bowel disease, and rheumatoid disease), metabolic (diabetes mellitus and osteoporosis), cardiovascular diseases and psychiatric disorders. Bone Regenerative Approach

Reviewer 2 Report

Comments and Suggestions for Authors

The reviewed paper presents a discussion on endodontic regeneration therapy, but several weaknesses and flaws can be identified across different sections.

The introduction provides a broad overview of the topic but lacks clarity in defining the specific gaps in knowledge and the objectives of the study. It assumes prior familiarity with regenerative dentistry concepts, which may obstruct for a broader expertise individuals. Additionally, the transition between general concepts and the rationale for the study is weak, making it difficult to see how the paper's focus fits within the existing body of literature.

The methods section lacks sufficient detail regarding experimental design, selection criteria, and statistical analyses, making it difficult to assess the reproducibility of the findings. T

here is an overreliance on previously published studies without specifying how the methodologies align with the research objectives.

Furthermore, it does not discuss potential biases or limitations in study selection, which could impact the validity of the results.

In the results section, data presentation is sometimes vague, with certain findings lacking quantitative evidence or statistical validation, which weakens the scientific rigor. The discussion of the findings is largely descriptive rather than analytical, with little critical engagement with conflicting evidence or alternative interpretations.

The discussion tends to be overly optimistic, emphasizing the potential of regenerative approaches without sufficiently addressing the challenges, risks, or limitations, such as immune rejection, long-term viability, and clinical feasibility.

The conclusion is somewhat generic and does not propose concrete future research directions or practical implications for clinical translation.

Regarding references, the paper cites a mix of recent and older literature but lacks a clear justification for selecting certain references.

Some key recent advancements in regenerative endodontics are either overlooked or not critically analyzed.

Comments on the Quality of English Language

Can be improved

Author Response

We sincerely appreciate Reviewer 2 for careful review of our manuscript and thoughtful comments. In response, we have provided the following answers.

Major revision
Comment: The reviewed paper presents a discussion on endodontic regeneration therapy, but several weaknesses and flaws can be identified across different sections.
The introduction provides a broad overview of the topic but lacks clarity in defining the specific gaps in knowledge and the objectives of the study. It assumes prior familiarity with regenerative dentistry concepts, which may obstruct for a broader expertise individuals. Additionally, the transition between general concepts and the rationale for the study is weak, making it difficult to see how the paper's focus fits within the existing body of literature.

ANSWER: Thank you for your insightful feedback. We have carefully revised the manuscript to clearly present the specific objectives of our study, which provide the rationale for regenerative therapy in endodontics.

Page 3, Line 69

This review provides a comprehensive overview of current regenerative procedures and fundamental concepts in endodontics, with a primary focus on stem cell biology in pulp and bone regeneration. Stem cell therapy for pulp regeneration has evolved into a clinically applicable procedure, while bone regeneration strategies using stem cells present a promising future approach for treating apical periodontitis associated with large bone defects, such as endo-perio lesions. The discovery of human bone-forming cells has further clarified the principles of bone regeneration therapy and paved the way for innovative treatments for severe bone loss. These advances improve bone healing and stability, which ultimately improves the long-term success of endodontic treatments. 

Building on these findings, this review examines regenerative strategies in dental pulp, particularly cell-based approaches (cell transplantation) using dental pulp stem cells (DPSCs). Additionally, vital pulp therapy, which aims to preserve pulp vitality and maintain physiological function, is discussed. This review also addresses cell-free regenerative approaches (cell homing) that do not require exogenous cell transplantation, as well as emerging technologies for whole-tooth regeneration. 

In apical periodontitis, the relationship to systemic inflammation and its impact on healing potential must be considered when assessing the need for bone regeneration. Moreover, strategies for bone regeneration, including the use of bone graft substitutes and tissue engineering technologies, are being actively explored. Bone-forming cell-based therapies, such as the use of skeletal stem cells and immature osteoblast-like cells, show promise for future applications. By integrating recent advances in both pulp and bone regeneration, this review provides a comprehensive perspective on the current state and future directions of regenerative endodontics. 

Comment: The methods section lacks sufficient detail regarding experimental design, selection criteria, and statistical analyses, making it difficult to assess the reproducibility of the findings. There is an overreliance on previously published studies without specifying how the methodologies align with the research objectives.
Furthermore, it does not discuss potential biases or limitations in study selection, which could impact on the validity of the results.

Answer: Thank you very much for highlighting this valuable information. In this review, we have focused solely on the clinical application of pulp regeneration therapy using cell-based transplantation methods. We identified five completed clinical trials that employed various types of stem cells. Additionally, we have specified the selection criteria for the articles in Table 5 on Page 8, Line 268, as follows.

Page 8, Line 273

Table 1 summarizes selected articles on pulp regeneration therapy applied in clinical trials. Since animal studies were also excluded from this review, the number of human clinical trials using stem cells was limited. Of the five studies reviewed, two were randomized controlled trials, while three were non-controlled clinical studies. The targeted teeth for pulp regeneration included both immature and mature teeth, with or without apical periodontitis. The intervention in these studies involved a cell-based regenerative approach using various types of stem cells, either autogenic or allogeneic. To ensure a consistent assessment of the results, we included studies with a follow-up period of at least one year. Treatment success was defined by the resolution of infection-related signs and symptoms, revascularization of the pulp, root development, and narrowing of the apical foramen.

The first two studies focused on immature teeth. The first clinical study [55], investigated the use of human exfoliated deciduous dental pulp stem cells (SHEDs) for the treatment of traumatic immature incisors with pulp necrosis. Pulp vitality and dentin formation were assessed after a 12-month follow-up period using laser Doppler flowmetry and CBCT analysis [55]. In the second study [56], allogeneic bone marrow-derived mesenchymal stromal cells were used to treat immature non-vital teeth. After a 12-month observation period, the healing and tooth sensitivity were evaluated using electric pulp testing and cold tests [56].

Three studies focused on pulp regeneration in mature teeth. The first study [57] investigated the use of autologous mobilized dental pulp stem cells with granulocyte-colony stimulating factors (G-CSF) and an atelocollagen sponge, which were implanted into mature single-root canals diagnosed with irreversible pulpitis. Pulp sensitivity and dentin formation were assessed after 12 months using electric pulp testing, MRI signal intensity, and CBCT. The second study[58], investigated the efficacy of allogeneic human umbilical cord mesenchymal stem cells (UC-MSCs) in combination with platelet-poor plasma in the treatment of mature teeth with pulp necrosis or apical periodontitis (n=18). Pulp vitality was evaluated using electric pulp testing and laser Doppler flowmetry after 12 months [58]. The final study [59], explored direct pulp tissue transplantation, a technique that involves the direct transplantation of dental pulp stem cells, bypassing complex in vitro procedures. In this study, six mature necrotic permanent teeth were treated with minced pulp tissue grafts following apical-induced bleeding. After a follow-up period ranging from 19 to 42 months, periapical lesions were assessed both clinically and radiographically, and pulp sensitivity was evaluated using electric pulp testing [59].

Comment: In the results section, data presentation is sometimes vague, with certain findings lacking quantitative evidence or statistical validation, which weakens the scientific rigor. The discussion of the findings is largely descriptive rather than analytical, with little critical engagement with conflicting evidence or alternative interpretations.

Answer: Thank you very much for pointing out the valuable information. We have summarized the data of five articles and added a summary.

Page 9, Line 307

The summary of the above five studies has shown promising results in both immature and mature teeth, demonstrating the potential for pulp healing and revitalization. Among the 70 teeth examined in the five studies, 65 were diagnosed with pulp necrosis or periodontitis. After a 12-month follow-up, all 65 teeth (100%) showed a reduction in periapical lesions indicating a high success incidence. The five teeth from Nakashima et al. (2017) were excluded due to the diagnosis of irreversible pulpitis. Regarding pulp vitality, of the 70 teeth tested, 64 (91.4%) responded positively to electric pulp testing, while six teeth (8.6%) showed no response. It is noteworthy that five of the non-responding teeth (7.1%) were from the study by Kim et al., which involved the direct transplantation of minced pulp tissue.

To further evaluate the viability of the pulp, two studies [55,58] utilized laser Doppler flowmetry and reported an increase in perfusion units between 6.39 and 17.5. In one study [57], the signal intensity of the regenerated pulp compared to the normal pulp was examined using MRI. It was found that all five teeth examined had a signal intensity comparable to that of normal teeth, suggesting complete pulp regeneration after 24 months. In the remaining two studies [59,56], neither MRI nor laser Doppler flowmetry was used to assess viability.

Regarding root dentin deposition and apex closure, a study on immature teeth by Gomez et al. reported that all 15 teeth demonstrated apical closure and dentin deposition. Xuan et al. observed a mean decrease in root length of 2.64 ± 0.17 mm over 12 months. Nakashima et al. reported dentin deposition in 3 out of 5 mature teeth. However, dentin deposition was not explicitly mentioned in the other two studies.

As far as the safety of stem cell implantation is concerned, urine and blood tests were carried out in two studies. These studies analyzed immune cell profiles, including CD4+ T lymphocytes, CD8+ T lymphocytes, B lymphocytes, and natural killer (NK) cells. In addition, key biochemical markers such as blood bilirubin, lactate dehydrogenase, creatine phosphokinase, uric acid, antistreptolysin O, C-reactive protein, rheumatoid factor, immunoglobulin A (IgA), IgG, IgM, C3, C4, and myocardial enzymes were within normal ranges, indicating that there was no systemic immune response or organ dysfunction following stem cell therapy. The remaining three studies did not mention specific laboratory assessments but reported no systemic or local adverse effects related to the procedure.

Comment: The discussion tends to be overly optimistic, emphasizing the potential of regenerative approaches without sufficiently addressing the challenges, risks, or limitations, such as immune rejection, long-term viability, and clinical feasibility.

Answer: Thank you very much for your thoughtful comment. We have updated the discussion to address the challenges, risks, and limitations of regenerative approaches.

Page 9, Line 337

Although stem cell-based therapy has been reported to be safe, allogeneic therapy remains susceptible to immune rejection. Since dental pulp stem cells (DPSCs) and mesenchymal stem cells (MSCs) are not immune-privileged, they can be recognized and rejected by both innate and adaptive immune responses [60]. This immune recognition may compromise treatment outcomes and lead to reduced longevity and viability of regenerated pulp tissue [61].

Furthermore, the American Association of Endodontists (AAE) (2021) and the European Society of Endodontology (ESE) (2016) [40, 62, 63] have not yet recommended stem cell-based pulp regeneration for routine clinical practice. Despite its adoption in clinical trials, several challenges hinder the feasibility of this therapy. These include difficulties in standardizing protocols for stem cell isolation and expansion, limited access to Good Manufacturing Practice (GMP) facilities, and high costs associated with production and application. Addressing these limitations is essential for the successful translation of pulp regeneration therapies into mainstream endodontic practice.

Comment: The conclusion is somewhat generic and does not propose concrete future research directions or practical implications for clinical translation.

Answer: We sincerely appreciate your insightful feedback. We have revised the conclusion to provide more specific future research directions and practical implications. Based on your suggestions, we have revised the conclusion section to enhance clarity and coherence.

Page 17, Line 566

Endodontic regeneration therapy holds significant potential for restoring dental pulp and bone while reducing the prevalence of chronic dental infections, such as apical periodontitis (AP). Both cell-based and cell-free approaches have shown promise in dental pulp regeneration, with studies reporting positive outcomes for both allogeneic and autologous stem cell transplantation. In addition, cell transplantation has demonstrated potential in mature teeth, suggesting that it may be a viable alternative in certain cases. However, long-term studies are still needed to fully evaluate the viability and efficacy of these approaches. Future research should focus on improving long-term stability, optimizing immunomodulation, and standardizing protocols for stem cell isolation and application. Although the efficacy of cell-based therapy for pulp regeneration has been demonstrated, the precise mechanisms by which the transplanted cells promote pulp regeneration and root development remain unclear. This could be due to either the facilitation of wound healing in the periapical tissue through immune tolerance or the regeneration of pulp tissue through tissue reorganization. To address this issue, it is crucial to identify suitable cells for pulp regeneration and determine whether autologous or allogeneic transplantation is more effective for reliable cell therapy. Further research and clinical trials are essential to addressing these limitations, with an emphasis on developing combination therapies and establishing cost-effective strategies.

Comment: Regarding references, the paper cites a mix of recent and older literature but lacks a clear justification for selecting certain references. Some key recent advancements in regenerative endodontics are either overlooked or not critically analyzed.

Answer: Thank you very much for pointing out areas for improvement and highlighting valuable information. We have incorporated details on recent advances in regenerative endodontics and added appropriate references.

Page 10, Line 351

Recent advances in regenerative technology include cell sheets, spheroids, organ-oids, and 3D bioprinting [61]. Cell sheets provide a scaffold-free structure that forms high density sheet structures out of cells and their extracellular matrix [62]. Spheroids are dense 3D cell aggregates with enhanced cell-cell and cell-matrix interactions which create a potent secretome that enhances angiogenesis and tissue repair [63, 64]. Dental pulp organoids have emerged as promising models by co-culturing DPSCs with endothelial cells in a 3D environment to create prevascularized organoids. Additionally, layered scaffolds, designed using computer-aided design (CAD), facilitate periodontal and dental tissue regeneration by incorporating stem cells, biomaterials, and growth factors [65, 66]. These technologies can help address challenges in stem cell therapy, such as poor cell retention and rapid cell death, while also advancing personalized dental treatments.

Reviewer 3 Report

Comments and Suggestions for Authors

Dear Authors,

I appreciate the opportunity to review your manuscript and commend you for the effort and dedication invested in this research. The chosen topic is highly relevant to modern dentistry, and your approach demonstrates a strong commitment to scientific advancement in the field. The primary objective of this work is to advance endodontic regenerative therapies as innovative solutions to overcome the limitations of conventional root canal treatments, focusing on restoring dental pulp functionality in pulpitis and regenerating bone defects caused by apical periodontitis to improve long-term tooth preservation and systemic health outcomes.

However, some areas could be improved to enhance the clarity and overall quality of the study.

The introduction does not clearly present the main objective of the work. Initially, some infectious problems are discussed, but the treatments described do not mention the importance of disinfection. I recommend including at least a brief discussion on the necessity of disinfection before using any regenerative material.

Additionally, it would be beneficial to clarify the criteria used in selecting the five tabulated articles. What is the relevance or relationship between their approaches? What distinguishes them from other reviewed studies? Providing these details will improve the reader’s understanding of the methodology employed.

Addressing these aspects will significantly strengthen the manuscript, ensuring that its findings and contributions are effectively communicated.

Best regards,

Author Response

We sincerely appreciate the careful review of our manuscript and your encouraging comments. In response to the feedback, we have provided the following answers.

COMMENT: The introduction does not clearly present the main objective of the work. Initially, some infectious problems are discussed, but the treatments described do not mention the importance of disinfection. I recommend including at least a brief discussion on the necessity of disinfection before using any regenerative material.

ANSWER: We sincerely appreciate you highlighting this important information. We have added the main objective of our study. We also have included a brief discussion regarding the disinfection topics.

Page 3, Line no 69

This review provides a comprehensive overview of current regenerative procedures and fundamental concepts in endodontics, with a primary focus on stem cell biology in pulp and bone regeneration. Stem cell therapy for pulp regeneration has evolved into a clinically applicable procedure, while bone regeneration strategies using stem cells present a promising future approach for treating apical periodontitis associated with large bone defects, such as endo-perio lesions. The discovery of human bone-forming cells has further clarified the principles of bone regeneration therapy and paved the way for innovative treatments for severe bone loss. These advances improve bone healing and stability, which ultimately improves the long-term success of endodontic treatments. 

Building on these findings, this review examines regenerative strategies in dental pulp, particularly cell-based approaches (cell transplantation) using dental pulp stem cells (DPSCs). Additionally, vital pulp therapy, which aims to preserve pulp vitality and maintain physiological function, is discussed. This review also addresses cell-free regenerative approaches (cell homing) that do not require exogenous cell transplantation, as well as emerging technologies for whole-tooth regeneration. 

In apical periodontitis, the relationship to systemic inflammation and its impact on healing potential must be considered when assessing the need for bone regeneration. Moreover, strategies for bone regeneration, including the use of bone graft substitutes and tissue engineering technologies, are being actively explored. Bone-forming cell-based therapies, such as the use of skeletal stem cells and immature osteoblast-like cells, show promise for future applications. By integrating recent advances in both pulp and bone regeneration, this review provides a comprehensive perspective on the current state and future directions of regenerative endodontics. 

Page 5, Line 162

In cell homing, effective disinfection of the root canal system and achieving a sterile environment are crucial. Tissue repair and regeneration can only take place in a sterile or highly disinfected microenvironment in which the host’s defense mechanisms do not trigger tissue-destroying proinflammatory processes [38]. In chronic infections, microorganisms penetrate the dentinal tubules and form biofilms. This infected dentin surface prevents cell adhesion and fails to provide structural support for mobilized cells in the main canals [39]. To improve disinfection, current protocols [40] recommend the use of a lower concentration of sodium hypochlorite (1.5%-3%), followed by saline or ethylenediaminetetraacetic acid (EDTA). Calcium hydroxide or a triple antibiotic paste (0.1 mg/mL) are recommended as intracanal medicaments. At the second appointment, scheduled 1 to 4 weeks later, 17% EDTA is recommended as an irrigant to further optimize the microenvironment for regeneration.

COMMENT: Additionally, it would be beneficial to clarify the criteria used in selecting the five tabulated articles. What is the relevance or relationship between their approaches? What distinguishes them from other reviewed studies? Providing these details will improve the reader’s understanding of the methodology employed. Addressing these aspects will significantly strengthen the manuscript, ensuring that its findings and contributions are effectively communicated.

ANSWER: We sincerely appreciate you highlighting this important information. We have added information regarding the tabulated articles.

Page 8, Line no 273,

Table 1 summarizes selected articles on pulp regeneration therapy applied in clinical trials. Since animal studies were also excluded from this review, the number of human clinical trials using stem cells was limited. Of the five studies reviewed, two were randomized controlled trials, while three were non-controlled clinical studies. The targeted teeth for pulp regeneration included both immature and mature teeth, with or without apical periodontitis. The intervention in these studies involved a cell-based regenerative approach using various types of stem cells, either autogenic or allogeneic. To ensure a consistent assessment of the results, we included studies with a follow-up period of at least one year. Treatment success was defined by the resolution of infection-related signs and symptoms, revascularization of the pulp, root development, and narrowing of the apical foramen.

The first two studies focused on immature teeth. The first clinical study [55], investigated the use of human exfoliated deciduous dental pulp stem cells (SHEDs) for the treatment of traumatic immature incisors with pulp necrosis. Pulp vitality and dentin formation were assessed after a 12-month follow-up period using laser Doppler flowmetry and CBCT analysis [55]. In the second study [56], allogeneic bone marrow-derived mesenchymal stromal cells were used to treat immature non-vital teeth. After a 12-month observation period, the healing and tooth sensitivity were evaluated using electric pulp testing and cold tests [56].

Three studies focused on pulp regeneration in mature teeth. The first study [57] investigated the use of autologous mobilized dental pulp stem cells with granulocyte colony-stimulating factors (G-CSF) and an atelocollagen sponge, which were implanted into mature single-root canals diagnosed with irreversible pulpitis. Pulp sensitivity and dentin formation were assessed after 12 months using electric pulp testing, MRI signal intensity, and CBCT. The second study[58], investigated the efficacy of allogeneic human umbilical cord mesenchymal stem cells (UC-MSCs) in combination with platelet-poor plasma in the treatment of mature teeth with pulp necrosis or apical periodontitis (n=18). Pulp vitality was evaluated using electric pulp testing and laser Doppler flowmetry after 12 months [58]. The final study [59], explored direct pulp tissue transplantation, a technique that involves the direct transplantation of dental pulp stem cells, bypassing complex in vitro procedures. In this study, six mature necrotic permanent teeth were treated with minced pulp tissue grafts following apical-induced bleeding. After a follow-up period ranging from 19 to 42 months, periapical lesions were assessed both clinically and radiographically, and pulp sensitivity was evaluated using electric pulp testing [59].

The summary of the above five studies has shown promising results in both immature and mature teeth, demonstrating the potential for pulp healing and revitalization. Among the 70 teeth examined in the five studies, 65 were diagnosed with pulp necrosis or periodontitis. After a 12-month follow-up, all 65 teeth (100%) showed a reduction in periapical lesions indicating a high success incidence. The five teeth from Nakashima et al. (2017) were excluded due to the diagnosis of irreversible pulpitis. Regarding pulp vitality, of the 70 teeth tested, 64 (91.4%) responded positively to electric pulp testing, while six teeth (8.6%) showed no response. It is noteworthy that five of the non-responding teeth (7.1%) were from the study by Kim et al., which involved the direct transplantation of minced pulp tissue.

To further evaluate the viability of the pulp, two studies utilized laser Doppler flowmetry and reported an increase in perfusion units between 6.39 and 17.5. In one study, the signal intensity of the regenerated pulp compared to the normal pulp was examined using MRI. It was found that all five teeth examined had a signal intensity comparable to that of normal teeth, suggesting complete pulp regeneration after 24 months. In the remaining two studies, neither MRI nor laser Doppler flowmetry was used to assess viability.

Regarding root dentin deposition and apex closure, a study on immature teeth by Gomez et al. reported that all 15 teeth demonstrated apical closure and dentin deposition. Xuan et al. observed a mean decrease in root length of 2.64 ± 0.17 mm over 12 months. Nakashima et al. reported dentin deposition in 3 out of 5 mature teeth. However, dentin deposition was not explicitly mentioned in the other two studies.

As far as the safety of stem cell implantation is concerned, urine and blood tests were carried out in two studies. These studies analyzed immune cell profiles, including CD4+ T lymphocytes, CD8+ T lymphocytes, B lymphocytes, and natural killer (NK) cells. In addition, key biochemical markers such as blood bilirubin, lactate dehydrogenase, creatine phosphokinase, uric acid, antistreptolysin O, C-reactive protein, rheumatoid factor, immunoglobulin A (IgA), IgG, IgM, C3, C4, and myocardial enzymes were within normal ranges, indicating that there was no systemic immune response or organ dysfunction following stem cell therapy. The remaining three studies did not mention specific laboratory assessments but reported no systemic or local adverse effects related to the procedure.

Round 2

Reviewer 2 Report

Comments and Suggestions for Authors

Congratulations to authors